# Mitigating Ammonia Volatilization without Compromising Yield and Quality of Rice through the Application of Controlled-Release, Phosphorus-Blended Fertilizers

**Sajjad Ahmad** [1,2,3,4]**, Muhammad Yousaf Nadeem** [1,2,3,4]**, Shen Gao** [1,2,3,4]**, Quanxin Li** [1,2,3,4]**, Weike Tao** [1,2,3,4]**, Weiwei Li** [1,2,3,4]**, Yanfeng Ding** [1,2,3,4] **and Ganghua Li** [1,2,3,4,*]

1    College of Agronomy, Nanjing Agricultural University, Nanjing 210095, China
2    Key Laboratory of Crop Physiology Ecology and Production Management, Ministry of Agriculture, Nanjing Agricultural University, Nanjing 210095, China
3    Jiangsu Collaborative Innovation Center for Modern Crop Production, Nanjing Agricultural University, Nanjing 210095, China
4    China-Kenya Belt and Road Joint Laboratory on Crop Molecular Biology, Nanjing 210095, China
*    Correspondence: lgh@njau.edu.cn; Tel.: +86-25-84396475; Fax: +86-25-84396302

**Abstract:** Ammonia ($NH_3$) volatilization from paddy fields is a major issue which leads to poor fertilizer use efficiency and is considered a severe threat to the atmosphere. The previous research studies gave importance to the use of nitrogen fertilizers to mitigate $NH_3$ volatilization, while very little emphasis was given to the role of other fertilizers, such as phosphorus (P), for the alleviation of $NH_3$ volatilization in rice fields. Considering P importance herein, we conducted two consecutive field experiments using an innovative, controlled-release, phosphorus-blended fertilizer (CRPBF, with levels CRP0, CRP1, and CRP2). We compared CP0 (in which no fertilizer was applied), CP1 (112.5 kg P ha$^{-1}$ P of locally recommended fertilizers), and CP2: (P and K blended fertilizers) to determine the best possible way to reduce $NH_3$ volatilization without affecting the yield and quality of rice. The results of the study suggested that the yield of rice increased significantly with the application of CRP1 (11.11 t ha$^{-1}$) and CRP2 (11.99 t ha$^{-1}$). The addition of CRP1 and CRP2 to the rice field also enhanced yield-related components, i.e., panicle weight, total spikelets per unit area, spikelets per panicle, and above-ground biomass. CRP0 showed a lower yield and related components when compared to CP2. The addition of CRP1 and CRP2 demonstrated lower protein contents when compared to other treatments. The CRPBF application improved starch content and taste scores, and reduced the chalkiness of the rice grain during both years. The results showed a decreasing trend in $NH_3$ volatilization from CRPBF amendments by improving the nitrogen use efficiency traits when compared to other treatments: CRP2, CRP1, and CRP0 reduced $NH_3$ volatilization by 45%, 35%, and 15%, respectively. The results of this study indicate that, due to the episodic nature of $NH_3$ volatilization, CRPBFs with 50% P and 100% P can markedly reduce $NH_3$ volatilization from paddy fields without compromising the yield and quality of the crop, and could be a promising alternative to the ordinary commercial fertilizers used in rice fields.

**Keywords:** controlled-release fertilizers; yield; quality; NUE; $NH_3$ volatilization

## 1. Introduction

Rice (*Oryza sativa* L.) is a staple food crop and, due to its high nutritive value, represents 50% of the world population's food [1]. The predicted exponential increase in population in the coming decades will exert a tremendous pressure on increasing rice productivity [2]. In an attempt to solve these problems, the rice scientific community has taken up the challenge of enhancing the productivity of rice through different agronomic practices and the use of fertilizers [3]. The use of fertilizers such as nitrogen (N) and P provides essential nutrients for rice growth and development. It is often the most limiting factor in the rice production

system [4]. However, excessive use of N in rice fields leads to ammonia ($NH_3$) volatilization due to the low use efficiency of N fertilizers in the fields [5]. The loss of N in the form of $NH_3$ volatilization not only increases production costs but also poses a major threat to the surrounding environment [6]. The reduction of $NH_3$ volatilization from the rice field through appropriate fertilizer management is crucial to addressing the situation [7,8].

The factors responsible for $NH_3$ volatilization from rice fields are a high pH and $NH_4$ concentration in the microsphere. This enhances urea hydrolysis, causing maximum $NH_3$ escape from the soil surface [9]. The loss of N could be controlled by fertilizer application strategies that reduce the soil pH in the microsphere [10]. The application of P fertilizers to paddy fields decreases the soil pH in the microsphere, which helps to reduce $NH_3$ loss [11]. In addition to this, the amendment of P fertilizers mobilizes calcium, which may slow down urea hydrolysis and, in turn, reduce $NH_3$ volatilization [5]. Some of the previous studies reported that the addition of P produces an acidic environment in the paddy soil; this is effective in inhibiting or reducing $NH_3$ volatilization [12]. P management is also important for improving rice yield and quality. P plays an important role in many physiological processes, viz., respiration, photosynthesis, cell division, and energy storage [13]. P is a structural constituent of several bio-chemicals, i.e., nucleic acids (RNA and DNA enzymes and its coenzymes). It encourages root growth and enhances the resistance of plants to diseases [13,14]. The addition of P to paddy soil improves the nutritional quality and eating quality of rice [15]. However, commercially available P fertilizers are easily fixed in soil. They also sometimes leach out with water through runoff and are therefore not available to the plants [16,17]. The global P demand increased from 41,151,000 tons in 2015 to 45,858,000 tons ($P_2O_5$) in 2020, mainly due to the low reported efficiency of P fertilizers, as plants take only 10% to 20% of P from the soil [18,19]. The currently available form of P needs improvement to enhance its bioavailability to crops [20].

Controlled-release fertilizers have been proven to improve the efficiency of N and P fertilizers due to their slow-release nature [21,22]. The gradual release of nutrients from controlled-release fertilizers saves fertilizer consumption, labor costs, and the time needed to grow rice because nutrients are available to the crop throughout its growing period, merely single-time application [23]. The use of controlled-release P fertilizers enhances crop yield by increasing fertilizer use efficiency [24]. The loss of N in the form of $NH_3$ from rice fields can be controlled through the application of controlled-release P fertilizers, as they reduce the pH of the paddy soil and maximum $NH_3$ loss occurs due to a high pH [9]. Some research studies observed that the use of blended urea and phosphate fertilizers reduced $NH_3$ volatilization due to a higher N uptake [5]. However, there are several limitations as more research was performed on controlled-release N fertilizers compared to controlled-release P fertilizers [25]. The slow-release characteristic of controlled-release P fertilizers is affected by many factors, such as temperature, humidity, organic acids secreted by crop roots, soil pH, and the soil's mineral composition [26]. Thus, the presently available controlled-release P fertilizers cannot fulfill the P demand of the crop during its critical growth period [27]. Hence, controlled-release P fertilizers needed to be further improved. In addition, their effect on $NH_3$ volatilization in rice production is also unclear.

In the current study, therefore, we use controlled-release, phosphorus-blended fertilizers (CRPBF) in rice fields for the first time. In this study, a novel CRPBF with different levels was tested to evaluate its effect on $NH_3$ volatilization, grain yield, and quality of rice. The objective of this study was to determine: I) whether CRPBF showed any significant effect on rice yield; II) if the CRPBF affected grain quality; and III) whether the CRPBF had any role in the reduction of $NH_3$ volatilization by improving nitrogen use efficiency.

## 2. Materials and Methods

### 2.1. Site and Design Description

On-farm trials were conducted with a randomized, complete block design, with three replications at Yangling town in Danyang city, located (31°54′31″ N, 119°28′21″ E) in Jiangsu Province, P.R. China, for two consecutive years (between 2019 and 2020). Our

preliminary survey indicated a subtropical climate with a yearly precipitation of 882 mm and a mean annual temperature of 16.4 °C (Figure 1). The soil of the experimental area was categorized as Orthic Acrisol (FAO soil taxonomy in 1974) (Table 1). The high-yielding rice cultivar Ninjing 8 was used as a test variety. Machine transplanting was performed with a transplanting density of 30 cm × 12 cm, using a rice transplanter (Jingguan ride-on, high-speed rice transplanter) in mid-June. Crop field management was consistent with local, conventional, high-yield management, mainly controlling diseases such as rice blast, sheath blight, and rice smut, as well as insect pests such as chilo suppressalis, borers, and rice plant hoppers. Field ridges were built in each experimental plot and protected with polythene sheets to separate the water and fertilizer for independent drainage and irrigation. In the present experiment, the CRPBF was a blend of coated P and coated potassium (K) fertilizers, in which P was used in different ratios. The source of nitrogen was urea, the phosphoruswas diammonium phosphate, and the potassium was potassium chloride. A one-time machine application was used for the controlled release of the P-blended fertilizers before transplantation. A comparison of all treatments was made with the CP1 treatment used as a reference treatment. The treatment structure is presented in Table 2.

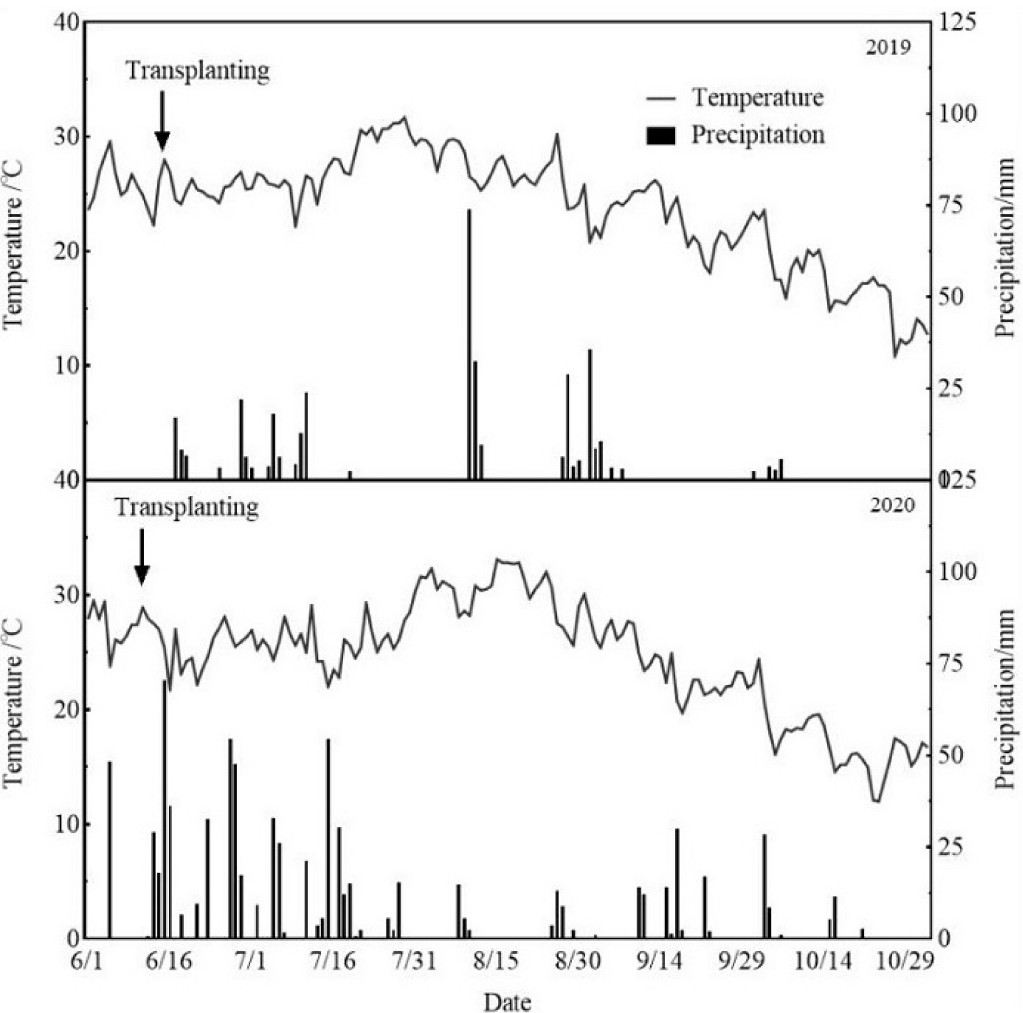

**Figure 1.** Daily temperature and precipitation during the rice-growing seasons in 2019 and 2020. Solid arrows show the time of rice transplanted.

**Table 1.** Main physical and chemical properties of the soil before transplanting.

| Soil Layer | pH | Organic Matter (gkg$^{-1}$) | Total N (gkg$^{-1}$) | NH$_4^+$—N (mgkg$^{-1}$) | NO$_3$—N (mgkg$^{-1}$) | Fast-Acting Olsen P (mgkg$^{-1}$) | Fast Acting NH$_4$ OAc-K (mgkg$^{-1}$) |
|---|---|---|---|---|---|---|---|
| 0–20 cm | 6.4 | 19.8 | 1.26 | 6.8 | 0.9 | 28.2 | 169 |

**Table 2.** Fertilizer application rates during both the rice-growing seasons 2019 and 2020.

| Treatments | Fertilizer Rate kg·ha$^{-1}$ | | |
|---|---|---|---|
| | **Nitrogen** | **Phosphorus** | **Potassium** |
| CP0 | 0 | 0 | 0 |
| CP1 | 225 | 112.5 | 180 |
| CP2 | 225 | 112.5 | 180 |
| CRP0 | 225 | 0 | 180 |
| CRP1 | 225 | 57 | 180 |
| CRP2 | 225 | 112.5 | 180 |

CP0: plot where no fertilizer was applied; CP1: 112.5 kg P ha$^{-1}$ P of locally recommended fertilizer; CP2: blend of P and K, 112.5 kg P ha$^{-1}$ and 180 kg K ha$^{-1}$; CRP0: no phosphorus fertilizer, 180 kg K ha$^{-1}$ potassium fertilizer with resin potassium chloride; CRP1: 57 kg P ha$^{-1}$, 50% reduced-resin phosphorus fertilizer, 180 kg ha$^{-1}$ potassium fertilizer with resin potassium chloride; and CRP2: 112.5 kg P ha$^{-1}$ resin phosphorus fertilizer, 180 kg K ha$^{-1}$ potassium fertilizer with resin potassium chloride. CRP0, CRP1, and CRP2 indicate the CRPBF levels.

## 2.2. Detail of Controlled-Release, Phosphorus-Blended Fertilizers

When compared to quick-acting fertilizers or controlled-release fertilizers, the one-time application of controlled-release fertilizers can reduce the number of fertilizations, save fertilizer consumption, and improve the yield and quality of rice. However, slow- and controlled-release fertilizers have problems, such as a nutrient release rate that is too fast in the early stage, which results in a nutrient release cycle that is too short or an insufficient early release. To address this problem, controlled-release fertilizers with different release rates are scientifically combined to form a controlled-release, blended fertilizer to achieve the synchronization of the nutrient release law with the crop demand law. The fertilizers used in our experiment were made by our lab by blending phosphorus and potassium fertilizers. Both fertilizers were coated with polyurethane material. The release pattern of the controlled-release, P-blended fertilizers is sigmoidal, and the release rate is 43%.

## 2.3. Sampling and Measurement

### 2.3.1. Crop Yield and Nitrogen Use Efficiency Traits

At physiological maturity, a 5 m$^2$ area in the center of each plot was harvested to determine the grain yield based on the standard moisture content of 13.5%. For the calculation of N use efficiency traits, a dried sample obtained from the above-ground biomass of each plot was cured at 105 °C for 30 min in an oven. The samples were then dried in an oven up to 80 °C until they reached a constant weight. The dried samples were ground into powder form with a plant grinder and passed through a 0.5 mm sieve. Finally, the total N uptake, plant N content, and total N accumulation were determined by the Kjeldahl N determination method [28]. The N recovery efficiency was calculated by the following formula:

$$RE_N = \frac{U_N - U_0}{F_N} \times 100\% \tag{1}$$

where $RE_N$ indicates recovery efficiency of N, $U_N$ represent the uptake of N (kg ha$^{-1}$) in the above-ground rice at the maturity stage in the N fertilized area and $U_0$ indicates the uptake of N (kg ha$^{-1}$) from the plots where no N fertilizer was applied, respectively, and $F_N$ represents the N application amount (kg ha$^{-1}$).

2.3.2. Grain Quality Measurement

The protein and starch content were found by milling a sample of whole rice into powder form using a pulverizer. The sample was then passed through a 100-mesh sieve and stored in a refrigerator at −20 °C. The protein and starch contents were determined later. The total starch content of the rice was measured by polarimetry. The RcTAIA model taste meter, which was developed by the Japanese Satake/Rlta10A Corporation, was used to determine the comprehensive taste value of rice. For the chalkiness %, randomly selected samples were placed under a spotlight and the area of chalky grain was determined. The chalkiness is expressed as the % of the total chalky area in the paddy sample to the total area of the sample, which is the product of the chalky grain rate and the chalky area.

2.3.3. $NH_3$ Volatilization Sampling and Measurement

The $NH_3$ volatilization was measured via the ventilation method in a closed chamber after transplantation [29]. Samples were first taken for three consecutive days, then taken every seven days. From the panicle initiation stage, the samples were taken at a gap of ten days until maturity [22]. $NH_3$ was collected using a pump and fed into the absorbent, which contained 2% boric acid mixed with an indicator composed of methyl red and bromocresol green. Samples were collected between 7:00 and 9:00 a.m. and between 1:00 and 3:00 p.m. over the entire rice-growing period [1]. The chemical reaction that occurred during the collection process, the reaction of boric acid and ammonia gas, is described as:

$$4H_3BO_3 + 2NH_3 = (NH_4)2B_4O_7 + 5H_2O.$$

The collected samples were titrated with a pre-determined acid solution to calculate the amount of trapped $NH_3$. The chemical reaction that occurred during the titration is as follows:

$$(NH_4)2B_4O_7 + H_2SO_4 + 5H_2O = (NH_4)2SO_4 + 4H_3BO_3.$$

The $NH_3$ volatilization flux (kg ha$^{-1}$ d$^{-1}$) is calculated by Equation (1):

$$AV = 2c(H_2SO_4) \times V(H_2SO_4) \times 10^{-3} \times M(NH_3) \times 10^{-3}/4 \times 24/\pi R2 \times 10,000 \qquad (2)$$

where $c(H_2SO_4)$ is the sulfuric acid concentration, (mol L$^{-1}$); $V(H_2SO_4)$ represents the volume of sulfuric acid consumed by titration (mL); $M(NH_3)$ is the molar mass of ammonia (g mol$^{-1}$); $R$ is the radius of the plexiglass cylinder (m), and 4 and 24 represent hours.

*2.4. Statistical Analysis of Data*

The statistical analysis was performed using statistix software 8.1 (Analytical Software, Tallahassee, FL, USA). The mean difference between the treatments was separated by the least significant difference (LSD) test at the probability level of 0.05. Figures were drawn using graph pad prism, version 8.0, R 3.6.1, and Microsoft Excel 2013.

**3. Results**

*3.1. Crop Yield and Yield-Related Components*

The end product of a crop is the yield and its related components; these determine the significance and productivity of the crop. The results of the study revealed that the CRPBFs with 100% P and 50% P demonstrated a significant increase in the rice yield, improving the rice yield from 11.11 t ha$^{-1}$ to 11.99 t ha$^{-1}$ during both rice-growing seasons (Table 3). A higher panicle weight was noted in the CRP2 treatment, and a lower panicle weight was indicated by the CP0 treatment. Among CRPBF treatments, CRP2 demonstrated higher panicle weights of 135.95 g and 134.11 g, whereas CRP0 perceived the lowest panicle weights of 114.01 g and 111.13 g, respectively. The addition of CRP2 demonstrated a maximum number of spikelets per unit area, while CP0 recorded a lower number of spikelets per unit area. Among CRPBFs, the CRP2 plot noted peak spikelet numbers per unit area, 56.68 and 45.73, and the lowest number of spikelets per unit area, 33.94 and 28.14,

was professed by the CRP0 treatment. Regarding spikelets per panicle, CRP2 recorded the maximum number of spikelets per panicle, whereas CP0 documented the minimum number of spikelets per panicle. Among CRPBFs, the greater number of spikelets per panicle, 128 and 125, was attained in the CRP2 plot, and the lower number of spikelets per panicle, 108 and 110, was observed in the CRP0 treatment. The maximum grain-filling percentage was recorded in the CRP2 treatments and the minimum grain-filling percentage was indicated in the CP0 plot. Among the CRPBF treatments, CRP2 revealed higher grain-filling percentages of 95.9% and 95.8%, while CRP0 determined lower grain-filling percentages of 94.8% and 94.6% during both years. CRP2 treatments recorded the maximum above-ground biomass when compared to other treatments. Among CRPBF treatments, CRP2 demonstrated greater above-ground biomasses of 20.54 t ha$^{-1}$ and 19.93 t ha$^{-1}$, while CRP0 indicated lower above-ground biomasses of 17.90 t ha$^{-1}$ and 17.11 t ha$^{-1}$ in both rice-growing seasons. In comparison to CP1 and CP2, the CRPBF with 0% P showed lower yield and yield-related components in both rice growing seasons.

**Table 3.** Yield and yield-related attributes under CRPBF treatment with different ratios of P during rice-growing seasons 2019 and 2020.

| Years | Treatments | Panicle Weight (g) | Total Spikelet's ($\times 10^3$ m$^2$) | Spikelets Per Panicle | Grain Filling (%) | Above Ground Biomass (t/ha) | Yield (t/ha) |
|---|---|---|---|---|---|---|---|
| 2019 | CP0 | 94.31 ± 2.97 d | 22.70 ± 0.23 d | 91 ± 0.33 d | 94.2 ± 0.12 | 13.9 ± 0.39 c | 7.30 ± 0.18 c |
| | CP1 | 114.91 ± 3.03 c | 34.17 ± 2.06 c | 112 ± 1.15 c | 95.2 ± 1.18 | 18.40 ± 0.25 ab | 10.02 ± 0.18 b |
| | CP2 | 119.91 ± 2.05 bc | 36.81 ± 1.07 c | 118 ± 1.76 b | 95.5 ± 0.42 | 18.57 ± 0.47 ab | 10.17 ± 0.13 b |
| | CRP0 | 114.01 ± 3.33 c | 33.94 ± 1.66 c | 108 ± 2.03 c | 94.8 ± 0.61 | 17.90 ± 0.32 b | 9.90 ± 0.10 b |
| | CRP1 | 127.99 ± 2.56 ab | 50.05 ± 0.73 b | 120 ± 1.15 b | 95.8 ± 0.64 | 19.90 ± 0.31 a | 11.71 ± 0.17 a |
| | CRP2 | 135.95 ± 2.04 a | 56.68 ± 2.42 a | 128 ± 1.73 a | 95.9 ± 0.57 | 20.54 ± 0.87 a | 11.99 ± 0.30 a |
| | *p*-value | 0.0001 | 0.0000 | 0.0000 | 0.1053 | 0.0001 | 0.0000 |
| 2020 | CP0 | 94.11 ± 1.79 c | 21.51 ± 1.39 d | 89 ± 0.58 d | 94.4 ± 0.25 | 13.17 ± 0.57 e | 7.27 ± 0.17 c |
| | CP1 | 113.98 ± 3.29 abc | 29.84 ± 1.89 c | 111 ± 1.14 c | 95.5 ± 0.21 | 17.99 ± 0.10 c | 9.99 ± 0.29 b |
| | CP2 | 117.90 ± 2.25 ab | 30.81 ± 2.13 c | 115 ± 1.57 b | 95.7 ± 0.57 | 18.43 ± 0.13 c | 10.12 ± 0.17 b |
| | CRP0 | 111.13 ± 2.48 bc | 28.14 ± 1.64 c | 110 ± 1.33 c | 94.6 ± 0.78 | 17.11 ± 0.32 d | 9.78 ± 0.11 b |
| | CRP1 | 125.19 ± 1.58 ab | 42.33 ± 1.30 b | 123 ± 0.58 a | 95.6 ± 0.60 | 19.20 ± 0.51 b | 11.11 ± 0.30 a |
| | CRP2 | 134.11 ± 1.25 a | 45.73 ± 1.21 a | 125 ± 1.89 a | 95.8 ± 0.41 | 19.93 ± 0.36 a | 11.67 ± 0.35 a |
| | *p*-value | 0.0277 | 0.0000 | 0.0000 | 0.750 | 0.0000 | 0.0000 |

CP0 indicates no fertilizer application, CP1 signifies local, regular fertilizer application, CP2 represent ordinary P- and K-blended fertilizers, CRP0, CRP1, and CRP2 indicate CRPBF. ± Standard error means (n = 3). Different letters in a single column indicate a significant difference between treatments at the 0.05 probability level.

### 3.2. Grain Quality Content

Significant variation was observed in grain quality contents, i.e., the total protein, total starch, chalkiness, and taste score (Table 4). Overall, the research study found a decrease in protein content in the CRP1 and CRP2 treatments when compared to the CRP0-, CP1-, and CP2-treated plots. Among the CRPBF treatments, the maximum protein contents, 8.49% and 8.38%, were noted in CRP0, whereas the lowest protein contents of 7.51% and 7.39% were recorded from the CRP2 treatment. The findings of the study reveal a higher starch content in the CRP1 and CRP2 treatments when compared to other treatments applied. Among CRPBFs, the highest starch contents of 76.81% and 76.41% was recorded from the CRP2-treated plot, while the lowest starch contents of 72.95% and 73.07% were revealed in the CRP0 treatment. Regarding chalkiness, the application of CRPBF demonstrated a positive effect on the chalkiness %. Maximum chalkiness values of 8.79% and 8.75% were witnessed by the CRP0 treatment, whereas minimum chalkiness values of 8.73% and 8.74% were noted by the CRP2 treatment. CRP1 and CRP2 treatment showed a significant effect on taste scores. With the application of CRPBF, higher taste scores of 64.67% and 64.47% were perceived for the CRP2-treated rice, whereas a lower taste score of 60.33% was noted for the CRP0-treated rice.

**Table 4.** Grain quality contents under CRPBF treatment with different ratios of P during rice-growing seasons in 2019 and 2020.

| Years | Treatments | Total Protein (%) | Total Starch (%) | Chalkiness (%) | Taste Score (%) |
|---|---|---|---|---|---|
| | CP0 | 7.01 ± 0.25 d | 70.63 ± 1.14 d | 7.99 ± 0.35 b | 59.00 ± 1.15 c |
| | CP1 | 8.33 ± 0.57 a | 75.81 ± 1.44 ab | 9.32 ± 0.31 a | 60.75 ± 1.44 bc |
| | CP2 | 8.15 ± 0.63 ab | 74.30 ± 3.37 bc | 9.12 ± 0.25 a | 61.31 ± 2.99 b |
| 2019 | CRP0 | 8.49 ± 0.11 a | 72.95 ± 1.06 c | 8.79 ± 0.16 a | 60.33 ± 2.02 bc |
| | CRP1 | 7.93 ± 0.57 b | 75.97 ± 2.61 a | 8.74 ± 0.11 a | 63.70 ± 1.73 a |
| | CRP2 | 7.51 ± 0.62 c | 76.81 ± 2.86 a | 8.73 ± 0.10 a | 64.67 ± 1.45 a |
| | *p*-value | 0000 | 0000 | 0.0149 | 0.0005 |
| | CP0 | 6.96 ± 0.27 c | 70.64 ± 1.57 c | 8.02 ± 0.25 b | 59.03 ± 1.12 c |
| | CP1 | 8.30 ± 0.16 a | 76.20 ± 1.86 a | 9.08 ± 0.09 a | 60.23 ± 1.24 c |
| | CP2 | 7.98 ± 0.52 ab | 74.97 ± 2.38 a | 9.03 ± 0.11 a | 61.73 ± 1.03 b |
| 2020 | CRP0 | 8.38 ± 0.31 a | 73.07 ± 1.37 b | 8.75 ± 0.18 a | 60.10 ± 1.15 c |
| | CRP1 | 7.60 ± 0.62 bc | 75.65 ± 1.99 a | 8.61 ± 0.15 a | 63.90 ± 1.55 a |
| | CRP2 | 7.39 ± 0.14 bc | 76.41 ± 1.98 a | 8.74 ± 0.11 a | 64.47 ± 1.09 a |
| | *p*-value | 0.0052 | 0.0002 | 0.0115 | 0.0000 |

CP0 indicates no fertilizer application, CP1 signifies local, regular fertilizer application, CP2 represent ordinary P- and K-blended fertilizers, and CRP0, CRP1, and CRP2 indicate CRPBF. ± Standard error means (n = 3). Different letters in a single column indicate a significant difference between treatments at the 0.05 probability level.

### 3.3. Nitrogen Concentration, Accumulation, and Use Efficiency Traits

N-related efficiencies consist of N uptake, translocation, utilization, and recovery, which is positively associated with the yield of paddy crops. The positive impact of CRPBF on the paddy soil enhanced the N-related attributes, such as total N absorption in the stem, leaves and grain, plant N content, N accumulation in spike, and N recovery efficiency, in both the years 2019 and 2020 (Table 5). When compared to CP1 and CP2, CRP0 treatments recorded less N-related attributes in both rice-growing seasons. Among CRPBFs, the CRP2 treatments enhanced the N-related contents and alleviated the total N uptake by 18% and 13%, plant N content by 39% and 27%, total P accumulation by 79% and 78%, and N recovery efficiency by 31% and 26%. This was followed by CRP1, which showed an increase in total N uptake by 15% and 10%, plant N content by 37% and 24%, total N accumulation by 67% and 55%, and N recovery efficiency was improved by 26% and 20% for each year, respectively.

**Table 5.** Nitrogen use efficiency traits under CRPBF treatment with different ratios of P during rice-growing seasons 2019 and 2020.

| Year | Treatments | Total N Uptake (kg/ha) | Plant N Content (%) | Total N Accumulation (kg/ha) | N Recovery Efficiency (%) |
|---|---|---|---|---|---|
| 2019 | CP0 | 171.32 ± 3.11 c | 0.92 ± 0.03 b | 16.11 ± 0.87 c | 23.34 ± 1.11 c |
| | CP1 | 203.88 ± 2.92 b | 1.19 ± 0.14 b | 22.07 ± 0.63 b | 38.82 ± 0.97 b |
| | CP2 | 208.66 ± 6.32 b | 1.60 ± 0.25 a | 25.12 ± 0.69 b | 40.08 ± 2.11 b |
| | CRP0 | 193.99 ± 3.50 b | 1.16 ± 0.03 b | 21.37 ± 0.17 b | 35.19 ± 1.17 b |
| | CRP1 | 233.71 ± 8.40 a | 1.63 ± 0.17 a | 36.83 ± 0.59 a | 48.43 ± 2.80 a |
| | CRP2 | 240.29 ± 2.29 a | 1.65 ± 0.01 a | 39.41 ± 0.54 a | 50.62 ± 0.76 a |
| | *p*-value | 0000 | 0.0041 | 0000 | 0001 |
| 2020 | CP0 | 150.12 ± 2.21 c | 0.81 ± 0.01 c | 15.13 ± 0.91 e | 21.34 ± 1.12 c |
| | CP1 | 177.92 ± 4.87 b | 1.17 ± 0.13 b | 21.47 ± 0.47 cd | 30.16 ± 1.62 b |
| | CP2 | 181.61 ± 1.35 b | 1.43 ± 0.17 ab | 24.71 ± 0.09 c | 31.06 ± 0.45 b |
| | CRP0 | 178.11 ± 0.29 b | 1.15 ± 0.12 b | 20.78 ± 0.47 d | 29.89 ± 0.10 b |
| | CRP1 | 194.68 ± 0.84 a | 1.46 ± 0.17 ab | 33.31 ± 0.13 b | 35.42 ± 0.28 a |
| | CRP2 | 200.39 ± 0.87 a | 1.49 ± 0.11 a | 38.26 ± 0.66 a | 37.32 ± 0.29 a |
| | *p*-value | 0000 | 0.0045 | 0000 | 00012 |

CP0 indicates no fertilizer application, CP1 signifies local, regular fertilizers application, CP2 represents ordinary P- and K-blended fertilizers, and CRP0, CRP1 and CRP2 indicate CRPBF. ± Standard error means (n = 3). Different letters in a single column indicate a significant difference between treatments at the 0.05 probability level.

### 3.4. Magnitude and Variation of Ammonia Volatilization

The variation in $NH_3$ volatilization and its loss from paddy fields under the application of controlled-release, phosphorus-blended fertilizers are presented in Figure 2. From the figure, it is clear that a rapid rise was seen in $NH_3$ volatilization after the application of fertilizers at initial stages during both rice-growing seasons (2019 and 2020). The $NH_3$ volatilization peak was observed in the first 10 to 20 days after transplantation. After that, the $NH_3$ volatilization trended downward, and the trend remained continuous until the end of the growing season during both years. In the year 2020, however, a minor peak was observed at 70 to 80 days after transplantation. For the fertilizer treatments, the highest $NH_3$ volatilization was observed for the local, high-yielding fertilizers (CP1), followed by the application of ordinary phosphate- and potassium-blended fertilizers (CP2), while the lowest $NH_3$ volatilization among all treatments was perceived from the plot where no fertilizers were applied (CP0) for both years. CRPBFs with different percentages of P showed a significant effect on $NH_3$ volatilization. The maximum $NH_3$ volatilization was observed from the CRPBF with 0% P (CRP0), followed by the CRPBF with 50% P (CRP1), whereas the lowest emission was observed from the CRPBF with a 100% P rate (CRP2). The CRP2 treatment mitigated $NH_3$ by 42% and 46%, followed by CRP1 at 34% and 37%, while the lowest volatilization mitigation percentages, observed by CRP0, were 14% and 16% for both years consecutively (2019 and 2020).

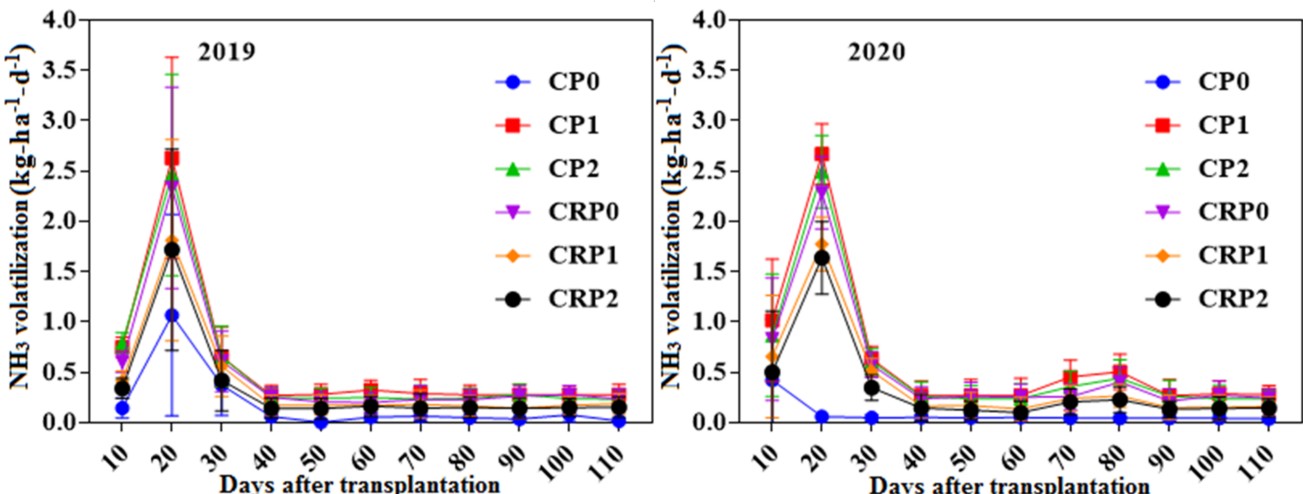

**Figure 2.** $NH_3$ volatilization under CRPBF treatments with different ratios of P during rice-growing seasons in 2019 and 2020. CP0 indicates no fertilizer application, CP1 signifies local, regular fertilizers, CP2 represents ordinary P- and K-blended fertilizers, and CRP0, CRP1, and CRP2 indicate CRPBFs. The vertical bars represent the standard deviation of the means.

### 3.5. Cumulative Ammonia Volatilization (CAV)

The cumulative ammonia volatilization (CAV) varied significantly from the paddy field with the application of different fertilizer treatments (Figure 3). In both the rice-growing seasons, the maximum CAV was recorded in the CP1 plot, followed by CP2, when compared to other treatments. Among CRPBFs, the highest CAV was recorded in the CRP0 treatment, whereas the CRP1 and CRP2 treatments noted the lowest CAV during both years (2019 and 2020). In both growing seasons, the CRP2 addition reduced the CAV by 61% and 53%, the CRP1 treatment mitigated the CAV by 49% and 42%, and the CRP0 amendment showed a 20% and 18% cut in the CAV for 2019 and 2020, respectively. The CAV of the 2020 rice-growing season was higher than that of the 2019 rice-growing season. The trend of two years of CAV with a proportion of N application rate is CP1 > CP2 > CRP0 > CRP1 > CRP2.

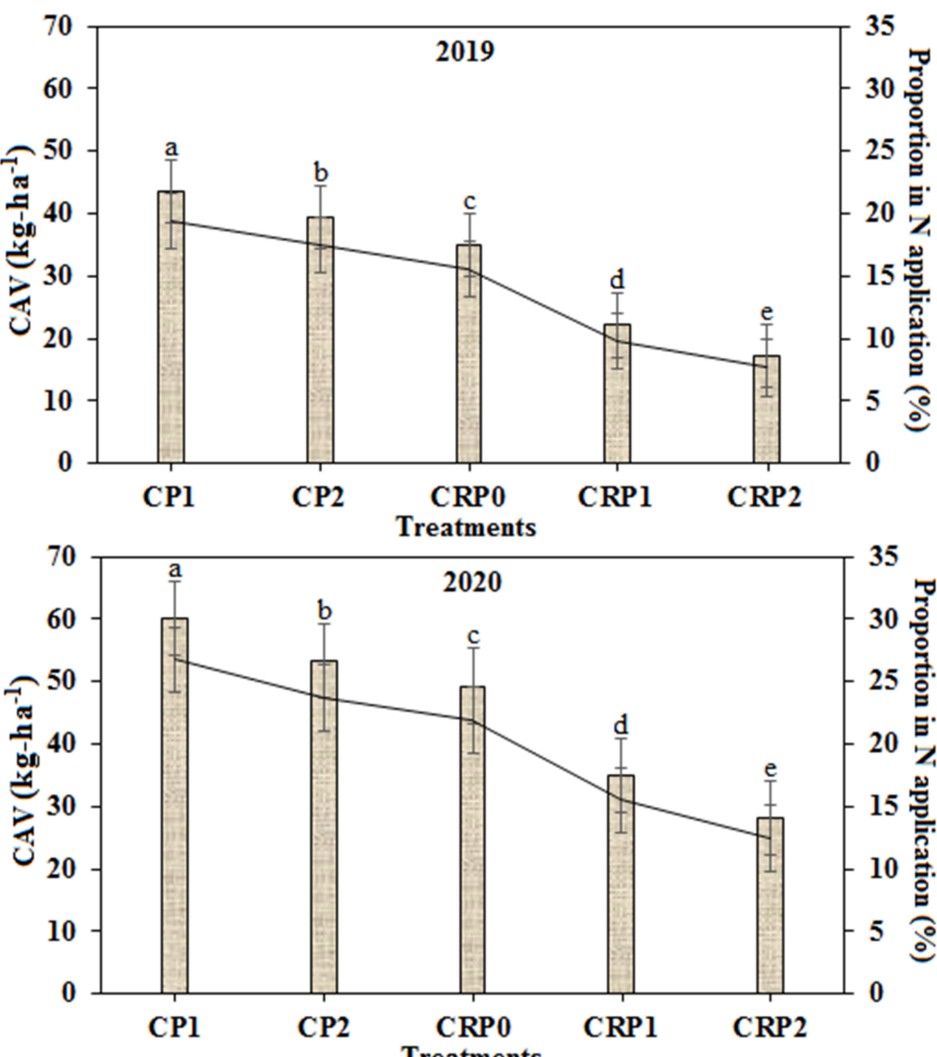

**Figure 3.** CAV: Cumulative ammonia volatilization under CRPBF treatment with different ratios of P during rice-growing seasons in 2019 and 2020. CP1 signifies local regular fertilizers, CP2 represents ordinary P- and K-blended fertilizers, and CRP0, CRP1, and CRP2 indicate CRPBFs. The vertical bars represent the standard deviation of the means. Different letters indicates a significant difference between treatments at the 0.05 probability level.

## 4. Discussion

### 4.1. The Effect of Controlled-Release, Phosphorus-Blended Fertilizers on Yield and Yield-Related Components

The application of fertilizers in an optimum quantity, of a proper source, and in a proper proportion is the key to sustainable crop production [30]. As phosphorus is a component of adenosine triphosphate, and due to its sufficient availability in the CRP1 and CRP1 treatments, it directly contributes to a large photosynthetic activity, which leads to luxuriant reproductive growth [31]. The results of this study revealed that the CRP1 and CRP2 treatments recorded a significant improvement in panicle weight when compared to other treatments (Table 3). This was due to the high availability of P in the CRP1 and CRP2 treatments, which stimulates plants growth, accelerates flowering, and helps to producing fertile panicles by enhancing the maximum uptake of nutrients from source to sink, leading to a high panicle weight [32]. Previous studies also observed that a maximum availability of P in the paddy soil enhanced the panicle weight by increasing the transfer of nutrients from source to sink [33]. Among yield components, spikelets per unit area was the basis of a stable, high yield of rice [34,35]. The increase in total spikelets per unit area was

linked with an effective number of ears. The addition of P to rice fields helps to produce a maximum number of effective ears [36]. A similar trend of P application enhancing spikelets per unit area due to a maximum number of effective ears was also reported in [37]. Regarding the alleviation of the above-ground biomass, CRPBFs with 50% P and 100% P recorded the maximum above-ground biomasses (Table 3). A possible reason for the increase in above-ground biomass could be correlated with a higher number of tillers [38]. The greater number of tillers produced eventually enhanced the above-ground biomass. Such an improvement in tillers number was due to an increased root biomass. The higher root biomass improved the nutrient uptake pattern by exploiting a greater volume of soil nutrients, which gave rise to an increase in the number of tillers [39]. The rice panicles and spikelet number is an of the important component of the harvest and is a considerable factor that affects the rice yield [40,41]. Our results showed that the number of spikelets per panicle was significantly increased with increasing grain yield levels (Table 3), which implies that the high yield of rice in our study could achieve a synchronous improvement in the spikelet number. The paddy yield showed a significant improvement with the addition of the CRP1 and CRP2 treatments (Table 3) as they made P less soluble, which was then readily available for a longer period to the crop during its growth period, helping to enhance the rice yield [42]. The increase in paddy yield due to the use of blended P fertilizers was also reported by [24]. Nalani et al. [42] that the addition of controlled-release P fertilizers enhanced the availability of P to the crop for a longer during its growth period, helping to enhance the paddy yield. The decrease in yield and its related attributes with the addition of the CRP0 treatment was due to the absence of P, which reduced the tillering capacity and retarded the root growth of crop [43].

### 4.2. Impact of Controlled-Release, Phosphorus-Blended Fertilizers on Grain Quality Characteristics

Most researchers found that protein and starch are the chemical components in rice grains which determine the quality of the rice [44]. Rice protein content is a complex quality index affected by the environment as well as by the fertilizer rate [45]. In general, it is believed that an increase in the fertilizer rate enhances the grain protein content. However, the amount, type, and time of application have a different effect on the protein content [46]. In the present research, a decrease in grain protein content was recorded in the CRP1 and CRP2 treatments when compared to the CP1 treatment. A possible reason for this might an inverse relation between the yield and protein content of grain; it was noted that the plots for which a maximum yield was recorded had lower grain protein contents. Eichi et al. [47] found a similar trend of a decreasing grain protein content with an increasing yield in wheat. In the present study, the starch content increased with the application of the CRP1 and CRP2 amendments. This mainly manifested due to an increase in the amylopectin content and a lower amylose content. An increase in amylopectin content with the application of CRP1 and CRP2 may be because the proper uptake of N improves branched starch enzymes, which enhances the amylopectin content while reducing the amylose content in grain [48–50]. The CRPBF treatments reduced the chalkiness %, which might be due to the loose accumulation of starch granules in rice grain [51]. It may also be due to the lower chalky-rice rate in CRPBF treatments, which was in line with [52]. The taste score of the rice is closely related to amylose and protein contents, which are negatively correlated with the taste value [53]. The results suggest a higher taste value in the CRPBF-treated rice because the treatments significantly reduced the chalkiness, protein, and amylose content of the rice. Our results are supported by [54].

### 4.3. Impact of Controlled-Release, Phosphorus-Blended Fertilizers on Nitrogen Use Efficiency Traits

The best N management is needed in rice fields to optimize crop growth while concurrently protecting the environment by reducing N losses through $NH_3$ volatilization [55]. The results of our study suggest that N losses can be significantly reduced through CRP1 and CRP2 amendments in rice fields, which considerably improve N use efficiency when

related to other treatments (Table 5). In the present study, the prolonged availability of P in the rhizosphere of rice plants due to application of the CRP1 and CRP2 treatments enhanced plant access to soil N, resulting in a better root system, improved N absorption, and improved N assimilation towards the stem and leaves, causing maximum absorption and an increase in plant N content. These results are consistent with previous studies, which concluded that N uptake is enhanced with the addition of P fertilizers [56,57]. The increase in the N accumulation rate in the current study was due to the remobilization of P with the application of CRP1 and CRP2 in the rhizosphere to improve the translocation of N from the rhizosphere to the vegetative part and then to the newly generated reproductive part, alleviating N accumulation. A similar result was also reported by [58], that P amendment to the soil enhances the translocation of N from the rhizosphere to the upper parts of the plants.

The recovery efficiency of N fertilizers depends not only on crop development but also on the management of the fertilizers [59]. The results of the experiment can be directly associated with different P rates in the soil. Fertilizer treatments such as CRPBF exerted a great impact on N recovery efficiency when compared to CP1 and CP2 treatment (Table 5). This indicates that a higher N recovery efficiency was assured by the longer availability of P in the soil due to the addition of CRP1 and CRP2. The maximum N recovery efficiency can also be achieved, as maximum P availability produces a better root system, leaves, and panicles that enhance the photosynthetic activity of the plant. In turn, this causes a good exploitation of N resources and achieves a good N recovery efficiency. The results are consistent with previous studies, which concluded that N recovery efficiency is enhanced with the addition of P fertilizers [57,60,61]. A considerable difference was recorded in the CRP0 treatment, as it perceived a lower N uptake, plant N content, plant N accumulation, and N recovery efficiency when compared to the treatments applied (Table 5). This was due to the weak growth of the rice population in the early growth stage, which affects N uptake and ultimately all other N-related indices [61,62]. Lower N-related indices due to the weak growth of rice at early stages were also documented by [63,64]. The result of our study showed that a moderate combination of N and P fertilizers enhances N uptake in the aboveground parts and enhances N recovery efficiency [4,65].

*4.4. Controlled-Release, Phosphorus-Blended Fertilizers Tradeoff with $NH_3$ Volatilization*

The $NH_3$ volatilization is regarded as the primary means of N loss in paddy fields, accounting for 9–40% of the total fertilizer applied [66,67]. The N loss from paddy fields closely relates to time and the types and ratios of applied fertilizers [68,69]. Factors that affect and control $NH_3$ volatilization in paddy fields include the soil $NH_4^+$-N, water sources (irrigation, precipitation), air temperature, wind speed, and the pH of the soil [70]. The higher $NH_3$ volatilization (Figure 2) immediately after transplantation was due to the urea applied, which rapidly hydrolyzed into $NH_4^+$-N and OH through the ammonification process as the hydrolysis rate is positively correlated with water availability, urease activity, pH, and air temperature [71,72]. As a result of fast hydrolysis, it enhances the $NH_4^+$-N concentration in the surface water and causes $NH_3$ volatilization [21]. At the early growth stage, seedling roots are not well established to uptake the proper rate of N from the soil, resulting in maximum $NH_3$ volatilization [73]. The second peak of $NH_3$ was in the middle of crop season in the year 2018. This may have been due to higher temperatures [70]. The reduction of $NH_3$ volatilization in the CRP1 and CRP2 treatment might be because the treatment reduced the rapid hydrolysis of urea and provided stability to $NH_4^+$-N over $NH_3$, thus reducing $NH_3$ volatilization [10]. Some previous studies observed that the blended application of P and N mobilizes Ca, which decreases the rate of urea hydrolysis [5]. The sparse canopy condition and lower N uptake in the CRP0 treatments promoted $NH_3$ volatilization [22,29]. Xia et al. [74] recorded the highest $NH_3$ volatilization rate due to a weak rice population.

## 5. Conclusions

This study provides a new perspective and suggests a strong potential for CRPBFs to reduce $NH_3$ volatilization from the rice paddy system without compromising the yield and quality of the paddy crop. Our research findings reveal that CRPBFs with 100% and 50% P significantly enhanced the yield and nitrogen use efficiency traits. However, CRPBFs with 0% P recorded lower yields and nitrogen use efficiency traits when compared to CP1 and CP2. Regarding grain quality characteristics, the CRPBFs with 100% P and 50% P enhanced starch content and taste value and reduced the chalkiness %. However, considering protein content, the CRPBF with 0% P provided better performance when compared to other treatments during both years, 2019 and 2020. In terms of $NH_3$ volatilization, the CRPBF addition seemed to efficiently alleviate $NH_3$ volatilization relative to other treatments applied. The results obtained from the present study support the role of CRPBF in reducing $NH_3$ volatilization from paddies and concurrently increasing the yield and quality of rice. Keeping these prominent features of CRPBF in mind, it is suggested that more research studies are required to explore the role of CRPBFs in a broader sense to effectively reduce $NH_3$ volatilization and maintain higher yields and quality in the paddy field.

**Author Contributions:** Conceptualization, G.L.; methodology, S.A.; software, M.Y.N., Q.L. and S.G.; validation, S.A. and M.Y.N.; formal analysis, S.A.; investigation S.A. and G.L.; resources, Y.D.; data curation, M.Y.N., Q.L., W.T. and W.L.; writing—original draft preparation, S.A. and G.L.; writing—review and editing, S.A., G.L., W.T. and W.L.; visualization, G.L.; supervision, G.L. and Y.D.; project administration, G.L. and Y.D.; funding acquisition, G.L. and Y.D. All authors have read and agreed to the published version of the manuscript.

**Funding:** This work was supported by the National Natural Science Foundation of China (31871573) and the Provincial key R & D plan, Modern Agriculture (BE2021361, BE2019377).

**Data Availability Statement:** Not applicable.

**Conflicts of Interest:** All the authors declare no conflict of interest.

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
