# Peer review of "Mitigating Ammonia Volatilization without Compromising Yield and Quality of Rice through the Application of Controlled-Release, Phosphorus-Blended Fertilizers"

_agronomy, doi:10.3390/agronomy13020448_

Round 1
Reviewer 1 Report
This paper presented a field study on the effects of addition of controlled-release phosphorus blended fertilizers (CRPBF) on rice yields, rice quality and NH3 volatilization from paddy field. The findings have important implications in multi-aspects including crop production, environmental protection and nitrogen management. I recommend this paper to be published in the Agronomy journal as soon as all my concerns are resolved.
Main concern:
The method of sampling and measurement of NH3 concentration needs to be described in more detail. Some important information, such as method detection limit, precision, of the titration method of determining NH3 should be given.
Other specific comments:
Line 99-130 (Section 2.1): In this study, there are up to six treatments with different fertilizers and application rates. Thus it is more concise and readable if the information of and differences among the treatments are presented in a table.
Line 141: Please check if the number “-1” in “FN-1” should be in superscript format. “UN” and “U0” have the similar problem. I also recommend that the units of UN, U0 ad FN-1 are specified.
Line 143: Please check if the number “-1” in “FN-1” should be deleted.
Line 156: What is “ventilation method”? Please give the reference related to this method or a brief description of it.
Line 163: What does 0.12 stand for? Please give the reference related to this method. Besides, please ensure the two sides of the equation have the same units.
Line 201-202 (Table 2): What does “Total spikelet’s” mean? If it means “total spikelets per unit area”, the unit (×103 m2) may not be correct. Beside, what does “Filled grain (%)” mean? Please give an accurate expression of the variable.
Line 204-205 (Table 2): The expressions such that “± Standard error mean (n=3). Significant at P<0.05; Significant at P<0.01; Significant at P<0.001” are hard to read and need to be rephrased. Similar issues exist in Line 204-205(Table 3), and Line 251-252(Table 4).
Line 207: Generally, a unit like percent sign (%) is not used in the headings in an article.
Line 211: Does the percent sign (%) in “protein content %” stand for “rate”? If so, I suggest changing it to something like “protein content rate” or “protein content rate (%)”, or “protein content (%)”, which in my opinion, are more formal expressions of a terminology.
Line 207-229 (Section 3.1 and 3.2): The texts in the two sections are somewhat lengthy and need to shorten.
Line 230-231: The units, percent signs (%), should be given in parentheses.
Line 240: The units (kg/hm2 and %) can be removed since they have been indicated in Table 4.
Line 272(Fig.2): In lines 157-158, the authors say the samples were collected on consecutive three days and then seven days befoe panicle initiatio stage and every ten days from then on. However, Fig. 2 only showed data with a ten-day interval.
Line 287: The VAC values of treatment CP0 could be presented in Fig. 3.
Line 296: The full name of the term “ATP” should be given.
Line 277-291: I suggest that the specific emission factors of N (“proportion in N application” in Fig. 3) for different treatments are also given in the text, because the information of NH3 emission factor are very important and useful in fields of ecology and atmospheric environment.
Author Response
Please see the attachment of revised paper below
Point-by-point response to reviewers
Dear Editor/ Reviewers
Thank you very much for supervising the review process of our manuscript and highlighting that our research article fits the journal and inviting us to re-submit the article to the journal. We found very valuable comments and recommendations from the reviewers. These suggestions/comments helped us to improve the overall quality of our manuscript. Following are the comment wise responses to the questions or comments of worthy reviewers. We have made our best attempt to address their valuable recommendations.
Note: All the changes are highlighted in red colour throughout the text and the response to the reviewers are highlighted in blue.
Reviewer comments and response
Comment # 1: The method of sampling and measurement of NH3 concentration needs to be described in more detail. Some important information, such as method detection limit, precision, of the titration method of determining NH3 should be given.
Thank you for your comments and valuable suggestion. We have described the sampling and measurement of NH3 concentration in detail as recommended. NH3 was collected using a pump and fed into the absorbent containg 2% boric acid mixed with an indicator composed of methyl red and bromocresol green; samples were collected 7:00-9:00 AM and 1:00-3:00 PM over the entire rice growing period The chemical reaction that occurs during the collection process: the reaction of boric acid and ammonia gas: 4H3BO3+2NH3 = (NH4)2B4O7+ 5H2O. The collected samples were titrated with pre-determined acid solution to calculate the amount of trapped NH3. The chemical reaction occur during the titiration are as under. (NH4)2B4O7+H2SO4+5H2O = (NH4)2SO4+4H3BO3.
Comment # 2: Line 99-130 (Section 2.1): In this study, there are up to six treatments with different fertilizers and application rates. Thus it is more concise and readable if the information of and differences among the treatments are presented in a table.
Thank you for the suggestion, the treatments have been presented in the table as advised.
Comment # 3: Line 141: Please check if the number “-1” in “FN-1” should be in superscript format. “UN” and “U0” have the similar problem. I also recommend that the units of UN, U0 ad FN-1 are specified.
Thank you for indicating the mistake. The text is revised and issue has been solved. The units of UN and U0 and FN were indicated in the material and method section. UN and U0 respectively represent the N uptake (kg ha-1) of the above-ground rice in the N-applying area and the non-nitrogen-applying area at the maturity stage and FN represents the N (kg ha-1) application amount.
Comment # 4: Line 143: Please check if the number “-1” in “FN-1” should be deleted.
Thanks, we re-write it as FN and removed FN-1 from the text has been removed.
Comment # 5: Line 156: What is “ventilation method”? Please give the reference related to this method or a brief description of it.
Thank you for pointing out the deficiency. The related reference has been added in the text.
Comment # 6: Line 163: What does 0.12 stand for? Please give the reference related to this method. Besides, please ensure the two sides of the equation have the same units.
Thank you very much for highlighting the major issue in the formula. The formula has been revised and explained briefly in the ammonia volatilization sampling and measurement section.
Comment # 7: Line 201-202 (Table 2): What does “Total spikelet’s” mean? If it means “total spikelets per unit area”, the unit (×103 m2) may not be correct. Besides, what does “Filled grain (%)” mean? Please give an accurate expression of the variable.
Thank you for your comment.. The total spikelets represent the number of spikelets per hectare, we measured area of 10m2 and it is multiply with 103 to represent the data for one hectare. Filled grains (%) is actually grain filling percentage and we have added accurate variable to the manuscript.
Comment # 8: Line 204-205 (Table 2): The expressions such that “± Standard error mean (n=3). Significant at P<0.05; Significant at P<0.01; Significant at P<0.001” are hard to read and need to be rephrased. Similar issues exist in Line 204-205(Table 3), and Line 251-252 (Table 4).
Thank you for the comments, the indicated text is revised as proposed in table 3 and table 4.
Comment # 9: Line 207: Generally, a unit like percent sign (%) is not used in the headings in an article.
Thank you very much for the suggesting the improvement. We have removed the (%)
Comment # 10: Line 211: Does the percent sign (%) in “protein content %” stand for “rate”? If so, I suggest changing it to something like “protein content rate” or “protein content rate (%)”, or “protein content (%)”, which in my opinion, are more formal expressions of a terminology.
Thanks for indicating the deficiencies, the texted has been revised as suggested and the (%) is removed.
Comment # 11: Line 207-229 (Section 3.1 and 3.2): The texts in the two sections are somewhat lengthy and need to shorten.
Thank you, the related texts has been shortened.
Comment # 12: Line 230-231: The units, percent signs (%), should be given in parentheses.
Thanks, the mistake is corrected in the manuscript as advised.
Comment # 13: Line 240: The units (kg/hm2 and %) can be removed since they have been indicated in Table 4.
Thank you, the suggested portion has been removed from the article.
Comment # 14: Line 272(Fig.2): In lines 157-158, the authors say the samples were collected on consecutive three days and then seven days before panicle initiation stage and every ten days from then on. However, Fig. 2 only showed data with a ten-day interval.
Thank you for the valuable comment, we have presented an average data of whole experiments from the transplantation to maturity stage at the ten days interval in the figure 2.
Comment # 15: Line 287: The VAC values of treatment CP0 could be presented in Fig. 3.
Thanks for the comment, the data related to CAV is not presented in the fig. 4. Because there is no fertilizers applied in the CP0 treatment.
Comment # 16: Line 296: The full name of the term “ATP” should be given.
Thanks for the comment. The full name of the abbreviation has been mentioned “Adenosine triphosphate”.

Reviewer 2 Report
The main goal of the study was to determine for the first time controlled-release phosphorus blended fertilizers (CRPBF) in rice field production. The Authors carried out the field experiment with randomized complete block design having in three replications at Yangling town in Danyang city located in Jiangsu Province, P.R. China for two consecutive vegetation seasons 2019-20. The Authors showed interesting results. The results obtained from the study supports the role of CRPBF to reduce NH3 volatilization from paddy and concurrently increases yield and quality of rice. Work is written correctly, research methods selected correctly as well, results presented in a clear manner, conclusions correspond to the assumed research goals, enough literature included, although it was found in a few cases the bad citation and spelling of the literature (pointed at the text). Please also use the correct spelling of units.
After all corrections the paper can be publish in the MDPI Agronomy Journal.

Author Response
Please see the revised paper below in the attachment
Point-by-point response to reviewers
Dear Editor/ Reviewers
Thank you very much for supervising the review process of our manuscript and highlighting that our research article fits the journal and inviting us to re-submit the article to the journal. We found very valuable comments and recommendations from the reviewers. These suggestions/comments helped us to improve the overall quality of our manuscript. Following are the comment wise responses to the questions or comments of worthy reviewers. We have made our best attempt to address their valuable recommendations.
Note: All the changes are highlighted in red colour throughout the text and the response to the reviewers are highlighted in blue.
Reviewer comments and response
Comment # 1: Change the topic from capital letters to small letters.
Thanks for your suggestion. We have change the font size of topic from capital to small letters.
Comment # 2: the unit t/ha is not popular. Please change to t ha-1.
Thanks, the suggested correction is incorporated. We write t ha-1 in the whole text instead of t/ha.
Comment # 3: Change the font size of corresponding author name and email.
Thanks for the comment, the font size of corresponding author name and email by making it to bold size.
Comment # 4: A scientific name "Oryza sativa" should be written in Italic.
Thanks for your comment, the scientific name of rice "Oryza sativa" is re-written in Itialics in the introduction section.
Comment # 5: "3" of the NH3 should be subscript. There are many NH3 without subscript.
Thanks for highlighting the mistakes.The subscripts were updated in the whole text as suggested and written as NH3 instead of NH3.
Comment # 6: The units (kg/hm2 and %) can be removed since they have been indicated in Table 4.
Thank you, the suggested portion has been removed from the article.
Comment # 7: [74] Reported in his" should be changed to "Xia et al. [74] reported in his"
Thanks for indicating the mistake, the text has been corrected.
Comment # 8: Change font size of refrence # 10.
Thanks for highlighting the mistake. The font size of refrence # 10 has been change according to the agronomy journal format.
Comment # 9: Put all the names of author in refrence # 42.
Thanks for the comment. We have added all authors name in the refrence # 42

Reviewer 3 Report
The manuscript entitled " MITIGATING AMMONIA VOLATILIZATION WITHOUT 3 COMPROMISING YIELD AND QUALITY OF RICE 4 THROUGH APPLICATION OF CONTROLLED-RELEASE PHOSPHORUS BLENDED FERTILIZERS" submitted to Agronomy is interesting and fits the scopes of the Journal. However, the characteristics of the Controlled-release phosphorous blended fertilizers are not explained. Please add the characteristics of the coat materials, the P release rate, and the type of P releases such as linear releases or sigmoidal releases et al. Moreover, the authors used a blend of coated P and coated K fertilizers. In this case the expression of Controlled-release phosphorous blended fertilizers is not suitable. "Blend fertilizer of coated P and coated K" may be better.
Introduction: Line 43; A scientific name "Oryza sativa" should be written in Ithalic.
Line 50: "3" of the NH3 should be subscript. There are many NH3 without subscript.
Line71-73: These sentences are not clear.
Line 77: This sentence is not clear. Why does the use of controlled-release fertilizers reduce energy consumption?
Materials and Methods: Please add the information on the controlled release P and K fertilizers.
Please add how and when were the fertilizers applied.
Line 141: FN - 1 is confusing like an equation. I think FN is enough.
Line 162: the unit hm-2 is not popular. Please change to m-2 or h-1.
Line 163: In equation (1) why do you multiply 0.12 at first?
The authors measured NH3 volatilization for 2h, so for the conversion to daily volatilization, the value should be multiplied by 12. Moreover, c is 2xM of H2SO4. M(NH3) should be 14 (N) not 17 (NH3), because the author converted to KgN, not Kg NH3. Please be careful to check the equation and the result data by this calculation. Please show the R (m).
Results
Line 239,240: Please explain about "total N absorption" and "total N accumulation". What is the base of them in shoot or in grain?
Figure 2.3: Please add replication number (N=) in the footnote.
Discussion
Line 324: "[42] Reported in his" should be changed to "Nalini Sharma et al.[42] reported in his"
Author Response

(The authors gave the same response as above.)
